# Prediction of the Damage Effect on Fiberglass-Reinforced Polymer Matrix Composites for Wind Turbine Blades

**DOI:** 10.3390/polym14071471

**Published:** 2022-04-04

**Authors:** Mariana Domnica Stanciu, Silviu Marian Nastac, Ionut Tesula

**Affiliations:** 1Faculty of Mechanical Engineering, Transilvania University of Brașov, B-dul Eroilor 29, 500360 Brașov, Romania; ionuttesula@yahoo.com; 2Mining and Metallurgical Section, Russian Academy of Natural Sciences, Sivtsev Vrazhek 29/16, 119002 Moscow, Russia; 3Faculty of Engineering and Agronomy, “Dunarea de Jos” University of Galati, 810017 Braila, Romania

**Keywords:** polymer matrix, composite, finite element analysis, delamination

## Abstract

The structure of wind turbine blades (WTBs) is characterized by complex geometry and materials that must resist various loading over a long period. Because of the components’ exposure to highly aggressive environmental conditions, the blade material suffers cracks, delamination, or even ruptures. The prediction of the damage effects on the mechanical behavior of WTBs, using finite element analysis, is very useful for design optimization, manufacturing processes, and for monitoring the health integrity of WTBs. This paper focuses on the sensitivity analysis of the effects of the delamination degree of fiberglass-reinforced polymer composites in the structure of wind turbine blades. Using finite element analysis, the composite was modeled as a laminated structure with five plies (0/45/90/45/0) and investigated regarding the stress states around the damaged areas. Thus, the normal and shear stresses corresponding to each element of delaminated areas were extracted from each ply of the composites. It was observed that the maximum values of normal and shear stresses occurred in relation to the orientation of the composite layer. Tensile stresses were developed along the WTB with maximum values in the upper and lower plies (Ply 1 and Ply 5), while the maximum tensile stresses were reached in the perpendicular direction (on the thickness of the composite), in the median area of the thickness, compared to the outer layers where compression stresses were obtained. Taking into account the delamination cases, there was a sinuous-type fluctuation of the shear stress distribution in relation to the thickness of the composite and the orientation of the layer.

## 1. Introduction

### 1.1. The Damage to Wind Turbine Blades

According to several studies, 40% of all global damage to wind turbines is due to mechanical damage and malfunction; 20% is due to lightning; 9% is caused by lightning fires, overheated bearings or sparks caused by gears at the time, and forced braking/deceleration; and 4% is due to extreme weather events (storms, hurricanes etc.) [1,2,3,4]. The operation of the whole assembly depends to a large extent on the structural integrity of the wind turbine blades. Wind turbine blades can be damaged in many ways, depending on the blade design and materials used. In some cases, these modes of damage may result in complete destruction of the blade or may only require repair of the blade. The causes of damage to the wind turbine blades, highlighted by [1], are as follows: geometric factors associated with buckling, large deformation, crushing or folding; material factors associated with plasticity, ductile/fracture, rupture or crack damage; initial manufacturing imperfections, such as initial deformation, residual stresses or production defects; temperature factors, such as low temperature associated with cold running and high temperature due to fires and explosions; dynamic factors (sensitivity to deformation, inertia effect, damage) associated with the impact pressure resulting from the explosion, fallen objects or similar; and age-related impairments, such as fatigue cracking. The analysis of the risks of the blades is closely related to the safety in operation, understanding by this both the possibilities of minimization and elimination of some defects, as well as the reserve of robustness to some unforeseen demands. Quality monitoring is subject to reliability, as the latter is ensured by checking processes and technological equipment, by performing rigorous control of the quality of materials, design, manufacture, and final tests [3]. Li et. al. 2014 [3] identified the areas with the highest risk of damage. Along the length of the blade, the greatest damage occurs at a distance of 1/3 of the length seen from the tip of the blade to the fixing area in the rotor (68–72% of defects) and at 2/3 of the length (30–32% of defects) [3].

### 1.2. The Mechanical Properties of Glass-Fiber-Reinforced Polymer Matrix Composites

The challenge for wind turbine blade manufacturers is to find constructive solutions that ensure large blade lengths, but sufficient rigidity (no deformation under load) and at the same time light weight (to rotate at a minimum wind speed of 4.5–5 m/s); made of materials resistant to wear, damage, and fatigue; and low cost. Thus, thinner blades can deform under the wind load reaching the tower, which could lead to damage to both the blade and the tower. The composite materials used must be rigid so as not to allow large deformation of the blades, but also light so that their mechanical capacity is not diminished by their own weight. However, lighter blades also have the advantage of lighter loads at the base, as well as those on the rest of the structure, which in turn reduce the total weight and costs. The occurrence of blade damage is conditioned by the simultaneous action of factors: tensile stresses that occur due to centrifugal force and bending moments produced by wind pressure; plastic deformations that develop as a result of the stress of the material in the plastic field, correlated with the aggressive environmental factors that lead to the degradation of the elastic properties of the material; variable loads which, in the case of the blade, are due both to the variable action of the wind and the cyclic stresses that occur during the rotation of the blades by the kinetic energy of the wind, vibrations, and even resonance phenomena [5,6,7,8,9]. Research has shown that the most common types of damage are cracks; delamination; and peeling of joints, cracks, or exfoliations [10,11,12,13,14,15]. The weak point of the composites used in the construction of the blades is that on the thickness of the composite the strength is ensured by the strength of the matrix. As presented in the literature, there are different sections and models of blade stiffening [16,17,18,19,20]. Some researchers [20] studied the problem of delamination of T-type joints between the cover and the stiffening elements inside the blade. According to other researchers [21,22,23,24,25,26,27], the delamination mechanism consists of separation of plies from each other under loading. The risks of delamination occurrence consist of increasing the failure area due to interlaminar normal and shear stresses, leading to the sudden collapse of the entire structure [28,29,30]. Being subjected to dynamic loading due to the wind speed variation and the rotation mechanism of the wind turbine blades, the failure rate of the WTB increases with the extension of the debonding or delamination areas in the composite structure. According to [31], the delamination or crack tends to increase in its own plane due to material constraints and weak interface between plies. Different theoretical and numerical approaches regarding the delamination criteria are indicated in [30,31,32,33,34]. Some of them are based on the assumption that delamination occurs in pure interlaminar tension (mode I), pure interlaminar sliding shear (mode II), and pure interlaminar scissoring shear (mode III), if the corresponding interlaminar stress component exceeds the maximum interfacial strength [11]. The mechanical properties of the materials used in the construction of wind turbine blades are determined experimentally on samples subjected to various mechanical, static, or dynamic stresses, under controlled environmental conditions or subjected to thermal, humidity, or chemical treatments. Considering different databases on the state of stresses and deformations, delamination processes, extensive fatigue, and changes in the mechanical behavior of glass-fiber-reinforced plastic (denoted GFRP) to aggressive environmental factors (humidity, temperature, ultraviolet radiation) in WTBs, it can be seen that the results between different materials and the loading conditions are not transferable, and thus an additional high experimental effort is required for each new project and material used [31,32,33,34]. The numerical approach has the role of supplementing the realization of experimental tests on the real structure, reducing the consumption of materials, production, and time.

### 1.3. The Investigation Methods

Due to the large dimensions of the wind turbine blades that would involve high production costs, the experimental results of the GFRP properties are applied in numerical analyses of the whole structure, following either the state of stresses and deformations of the structure, or the delamination phenomenon, or the modal response of the structure, depending on the objective pursued. Thus, [35] presented a combined method, numerical and experimental, using the machine learning method in order to predict damage caused by delamination of the composite using modal analysis and numerical simulation. Ref. [36] focused on reverse-engineering of the process modeling of composite materials for wind-turbine rotor blades, starting from the real model and experimental measurements and applying the data in the numerical model of temperature distribution in thick laminates in the construction of the rotor. The main criteria for composite failure are based on the classical laminated plate theory (CLT), which assumes that each layer of the laminate is orthotropic and homogenous; the laminate is thin compared to the side dimension and for this reason Timoshenko’s theory can be applied. All displacements are small compared to the thickness of the laminates, the transverse share strains being negligible [37,38,39]. The Tsai–Wu and Tsai–Hill criteria are based on different strengths in tension and compression, and the mathematical modeling of different failure criteria is indicated in [38,39,40,41,42,43,44,45,46,47].

This paper focuses on the quantitative and qualitative effects of delamination severity in the polymer matrix of the wind turbine blade on its static behavior, using finite element analysis (FEA). Compared to other research, the novelty of this study is the post-effect analysis of the airfoil damage on stress and strain, in different degrees of degradation, with two areas of interest being developed. Future studies will address the experimental aspects of the entire structure of the wind turbine blade, starting from the simulations presented in the current research. Moreover, the FEA carried out in this research highlights the evolution of the magnitude of normal and tangential stresses, as well as the deformations in each layer of the composite depending on the direction of loading and the orientation of the layer.

## 2. Modeling of Damaged Fiberglass-Reinforced Polymer Matrix Composites from WTB Structures

### 2.1. The Geometrical Model of a WTB

A 1.5 m long WTB consists of two components: the airfoil that can be considered a thin-walled structure and the reinforcement I profile, both of which are made of fiberglass-reinforced composite. The geometrical model was designed in the Catia R19 solid modeling program. The WTB composite was modeled as a multilayer structure comprising five layers with the orientation of the RT500 plain weave fabric 0/45/90/45/0. Similar to the actual model, which consists mainly of ±45 layers, plus a small amount of randomly oriented fibers, gelcoat, and filling resin and with the inner I-profile fixed on the aerodynamic shell, the numerical model contains the same structure. From a mechanical point of view, the WTB is essentially a cantilever beam mounted on a rotating hub, and the aerodynamic shape of the blade consists of a relatively thin-walled structure, namely the outer shells (Figure 1). For the finite element analysis, ABAQUS Version 6.14, a commercial version for general-purpose finite element analysis, was used. Two types of WTB structures were investigated in order to determine the static responses: without delamination and with delamination (different degrees of damage).

Four cases were analyzed: case 1—blade without damage; case 2—single-layer damaged blade, in both areas; case 3—blade with two damaged layers, in both areas; case 4—wind turbine blade with three delaminated layers, in both areas (Figure 2). Three cases of delamination of the wind turbine blade were modeled and simulated, applied in two areas (area 1 and area 2), as can be seen in Figure 2. In the first case, the damage of one layer of the composite in both areas was considered. In the second case, the delamination of two layers was simulated, and in the third case, the deterioration of three layers was taken into account in the finite element analysis. The delaminated surfaces were determined taking into account the risk areas presented by the blades during operation according to the literature [2,3,4,5].

### 2.2. Preprocessing Steps of the Finite Element Analysis (FEA)

The WTB structure without and with damage was analyzed to determine the stresses and strain state. Two types of finite element meshing were used:Due to the complex geometry of the WTB airfoil, the structure was meshed using the incompatible mode eight-node bricks, denoted C3D8I, which are continuum elements with incompatible modes. This type of element assures good meshing at the contact between the inner surfaces of the airfoil. The C3D8I elements are first-order elements, being enhanced by incompatible modes to improve their bending behavior. The main effect of these modes is to eliminate the parasitic shear stresses that cause the response of the regular first-order displacement elements to be too stiff in bending [37,38].For the I-profile surface, C3D10M quadratic tetrahedral elements were used in order to solve the large-deformation problems and contact problems. This kind of element performs either the traditional node-to-surface or surface-to-surface contact meshing and exhibits minimal shear and volumetric locking.

The load of 600 N was applied on the WTB tip, in the y direction (Figure 2). Based on previous studies [14,17,18], the structure in the current research was subjected to a load perpendicular to the longitudinal axis of the blade, which has the greatest effect on the stresses and deformations of the structure. The displacements and rotations in all directions were constrained in the hub area, the WTB structure being subjected to bending. In a simple theory, the analyzed structure is the case of a cantilever beam subjected to bending. In addition to the normal bending stresses, shear stresses also appear in the composite layers.

Knowing that the anisotropic structure of the composites influences both the mechanical and non-mechanical characteristics, an anisotropy test of the wind turbine blade composite structure was performed by using the ultrasonic investigation method. It was possible to identify the degree of homogeneity/anisotropy of the composites used in the construction of the blade by determining the propagation time of the ultrasound at different points of the material established according to the polar coordinates. The investigation method is presented in detail in [47]. It can be seen in Figure 3 that the composite structure of the airfoil without damage is relatively homogeneous, especially in the three main directions of layer orientation.

The physical and elastic properties of the composite used in the simulation were determined experimentally, being presented in previous research papers [13,17,32,39]. Table 1 summarizes the elastic and mechanical properties used in the numerical analysis.

### 2.3. Postprocessing Steps of the Finite Element Analysis (FEA)

Two types of analysis were performed: static analysis and modal analysis. From the static analysis, the normal stresses, shear stresses, and displacements in the delaminated areas were extracted, being compared with the values obtained for the structure without defects. As the objective of the study focuses on the variation of stresses and strains in the delaminated areas, the nodes and elements that border these areas have been identified (Figure 4). The methodology of data analysis consists of collecting the normal stresses in two-dimensional directions, S11 and S22, and the shear stress S12 for each element in the considered area of each ply, in all four cases.

## 3. Results and Discussion

### 3.1. Static Analysis of the WTB—Stress State

For all analyzed cases, the distribution of maximum stress is observed on the opposite side of the force application, in the WTB hub area, regardless of the degree of blade damage. In general, a higher stress concentration develops in the transition region of the root-blade, as can be seen in Figure 5. The results of the static analysis show that the distributions of the normal stresses S11 and S22 and the shear stress S12 differ from one layer to another both in intensity and in orientation. Since the purpose of the study is the comparative analysis of the stresses in the damaged areas, the following results focus on the variation of the extracted stresses for each element that delimits the delaminated areas in each layer.

The delamination of the layers of the WTB structure has the effect of increasing the stresses in the delaminated areas by 30% (in the case of a single damaged layer) and up to 60% in the case of the delamination of two layers (third and fourth), in area 2. The stresses extracted from the second studied area of the wind turbine blade (towards the top of the blade—area 2) showed a higher increase than those from area 1, by 35%. This proves that an increase in the degree of blade damage, even over a relatively small area (2000 mm^2^), causes amplification of stresses in the delaminated area, generating the occurrence of stress concentrators and, according to different studies [27,28,29,30,40,41,42], the crack propagation occurs with an amplification factor that depends on the intensity of the time-varying loading, aggressive environmental conditions, vibrations, etc. Within this part of the paper, the results regarding the stresses of different WTB integrity states are presented. It can be observed that the delamination of the three composite layers increases the normal stresses in S11 and S22 in the outer layers of Ply 1 and Ply 5. Figure 6 and Figure 7 show a comparison between the variation of normal and tangential stresses in the analyzed cases, for each layer, in relation to the orientation of the composite layers. The envelope of the stresses in the elements at the boundary of the delaminated area was represented graphically using the values of the stresses extracted from each finite element, according to the numbering indicated in Figure 4. The maximum values of stresses in each layer depend on the orientation of the composite layers. This is observed in the geometric shape of the tension envelope. The more damaged the composite, the more non-linear the distribution of S11 stresses around the delaminated area, as can be seen for the middle layers (Figure 6 and Figure 7).

Considering that in the blade design, direction 1 coincides with direction x, and direction 2 coincides with direction z, according to Figure 4a, it can be seen that the maximum tension S11 of layer 1 coincides with the direction of the fabric orientation (direction x). The second layer, oriented at 45 degrees, has the maximum tensile stresses in the direction of the fibers. In the case of layer 3, located in the middle area of the airfoil thickness, the stresses are maximum in direction 1 (x), the direction that coincides with the direction of the fiber orientation (Figure 6 and Figure 7a). In the case of the normal stress S22, it can be observed that for layers 1 and 5, oriented parallel to the direction of the longitudinal axis of the blade, the stresses are maximum in the longitudinal direction, compared to layers 2 and 4, oriented at 45 degrees, alternating with the longitudinal axis. Maximums are oriented perpendicular to the longitudinal axis. As a result of the bending of the blade produced by the force applied to the tip of the blade, the tangential stresses S12 show an evolution dependent on the degree of delamination, the highest tangential stresses developing in case 4, as can be seen in Figure 6c and Figure 7c. An in-plane shear deformation causes an additional rotation of the fibers. Depending on the layup of a unidirectional laminate, high shear strains can occur.

The stress distribution profiles, according to the loading direction, are expressed in Figure 8. Along the blade, the tensile stresses are developed with maximum values in the upper and lower plies (Ply 1 and Ply 5), as indicated in Figure 8a,b. In the perpendicular direction, on the thickness of the composite, the maximum of the tensile stresses is reached in the median area of the thickness, compared to the outer layers where compression stresses are obtained (Figure 8c,d). Taking into account the delaminated cases, the shear stress distribution profiles are displayed in Figure 8e,f. As can be seen in Figure 8e,f, there is a sinuous-type fluctuation of the shear stress distribution relative to the thickness of the composite and the orientation of the layer. These results are in good agreement with the mathematical models developed by [45,46,47]. The magnitude of the stresses obtained in the four cases of delamination does not exceed the limit of rupture of the composite, which was determined experimentally by the authors in their previous studies [27]. In [31], a similar wind turbine blade was investigated experimentally. Although the experimental study aimed to determine the states of stress and strain in the area near the hub, the values obtained are comparable to those determined with FEA in the study presented, taking into account the differences given by the position of the measurement points. Thus, in case 1 (structurally integral blade), in the area near the top, the maximum stress is 17 MPa, and in the next area, 19 MPa. Experimentally, the tension in the hub area was 48 MPa. Taking into account the geometry of the blade, the evolution of the stress values in the case of FEA agrees with the values obtained experimentally by Iftimie et al. [31]. According to [36], several effects, including asymmetry of the blade sections, non-coincidence of the center of mass with the shear center, polarization between the aerodynamic and shear centers, offset of the center from the shear center, inclined installation of the blade, setting angle, cone angle, structural damping, gravitational loading, and aerodynamic loading influence the static and dynamic responses of WTBs. Compared to uniform blades, the asymmetry of the blade sections results in additional terms of stiffness, inertial forces, static displacements, damping terms, and excitation forces for nonlinear vibration of non-uniform blades. Thus, the delamination of the composite changes the relative equilibrium between all factors (the static displacements, gyroscopic forces, and elastic restoring forces) [48,49].

### 3.2. Effects of Delamination on the Displacement of WTBs

The distribution maps of the overall displacements of the wind turbine blade are similar for all four cases analyzed, being maximum at the tip of the blade (Figure 9). Similar to the variation in the stresses relative to the delamination cases, the overall displacement increases with increasing the WTB damage degree. Therefore, the displacement increases by 1.16% in case 4 (Table 2).

## 4. Conclusions

The present study focused on the effects of the delamination severity in fiberglass-reinforced composites used for wind turbine blade construction. The geometrical model was designed in Catia and imported in Abaqus, where all parameters related to the composite material structure (elastic properties, thickness of the layers, number of layers, etc.) were indicated according to the technological specifications of the composite manufacturers and previous experimental tests.

The main conclusions regarding the effect of delamination of the composite layers of the wind turbine blade on the stress states, using FEA, are as follows:In this study, different cases of delamination severity of the layers in different areas were assumed and the tensions in the damaged areas were analyzed.The stress state is influenced by the delamination severity, even if its surface is relatively small.The maximum values of normal and shear stresses occur in relation to the orientation of the composite layer.Even if the geometric stiffness remains unchanged in the case of a small delamination area, the crack propagation increases the damage during operation.

The numerical shapes of the isoprobable variation curves of the stresses analyzed in this study were limited to the case of static bending stress. The real structure is subjected to variable loads that induce a poly-axial state of tension, directed in three orthogonal directions, more precisely three-axial. As a result, the problem is complicated by the need to consider the effect of voltage concentration, average voltage, and non-synchronization of voltage cycles after the three directions of stress. Because fatigue degradation nucleates at the surface of the structure, where one of the three normal stresses is zero, fatigue in the three-axial stress state is reduced to a two-axial stress state. Numerical analysis showed increases in two-axial stresses with increasing material damage.

## Figures and Tables

**Figure 1 polymers-14-01471-f001:**
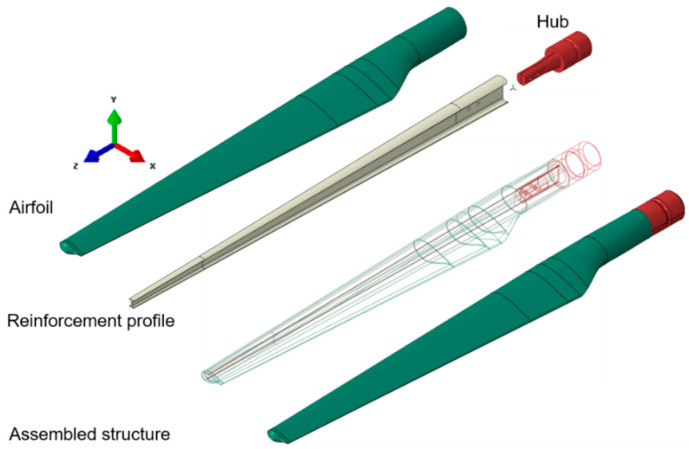
The main parts of the WTB.

**Figure 2 polymers-14-01471-f002:**
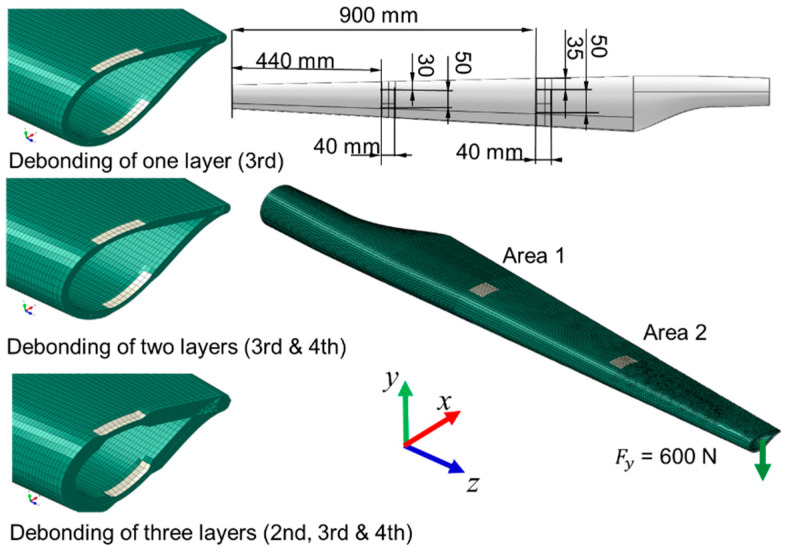
The structure of the wind turbine with the three cases of delamination (**left**—the types of delamination; **right**—the position of areas with delamination).

**Figure 3 polymers-14-01471-f003:**
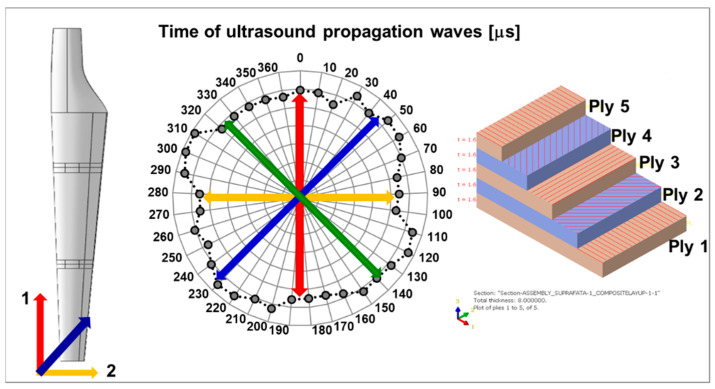
The variation of ultrasound propagation time in WTB composites in relation to the direction of composite layer orientation.

**Figure 4 polymers-14-01471-f004:**
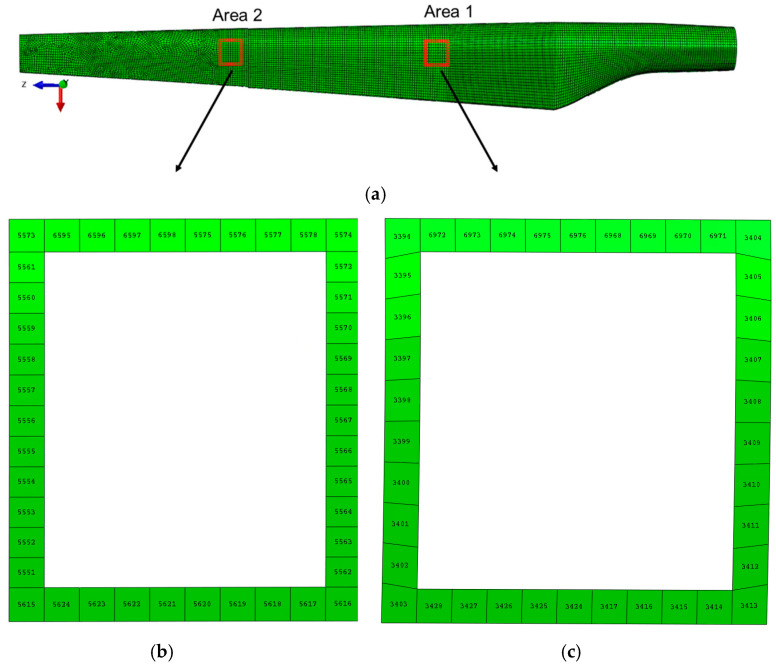
The elements on the contour of the delamination in the two areas: (**a**) the hypothetical two delaminated areas meshed in finite elements; (**b**) numbering of finite elements in the second delamination area; (**c**) numbering of finite elements in the first delamination area.

**Figure 5 polymers-14-01471-f005:**
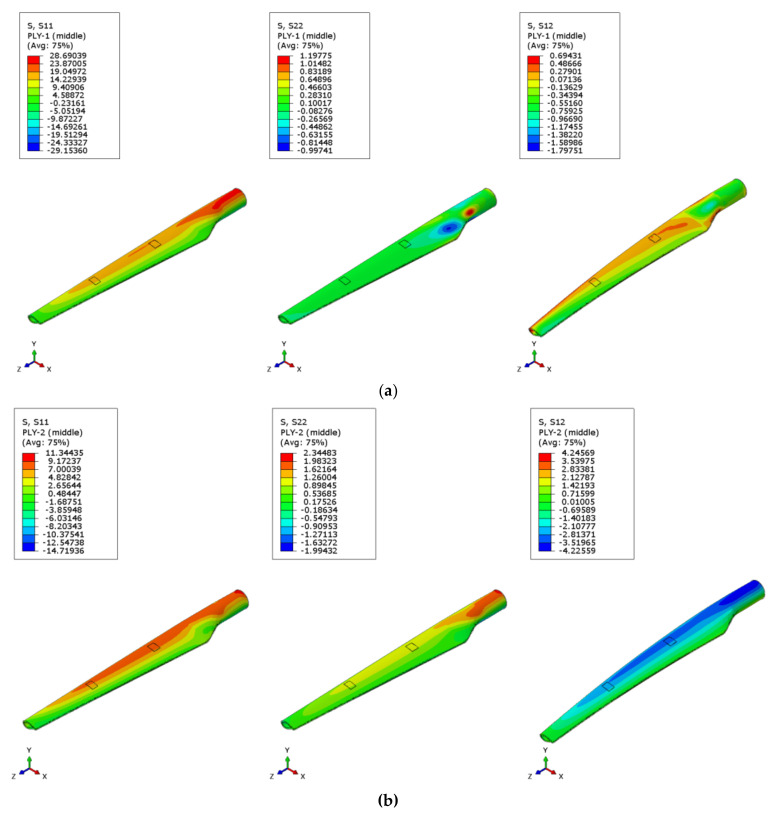
The distribution of normal and shear stresses in the WTB, without damage (case 1): (**a**) Ply 1; (**b**) Ply 2; (**c**) Ply 3; (**d**) Ply 4; (**e**) Ply 5.

**Figure 6 polymers-14-01471-f006:**
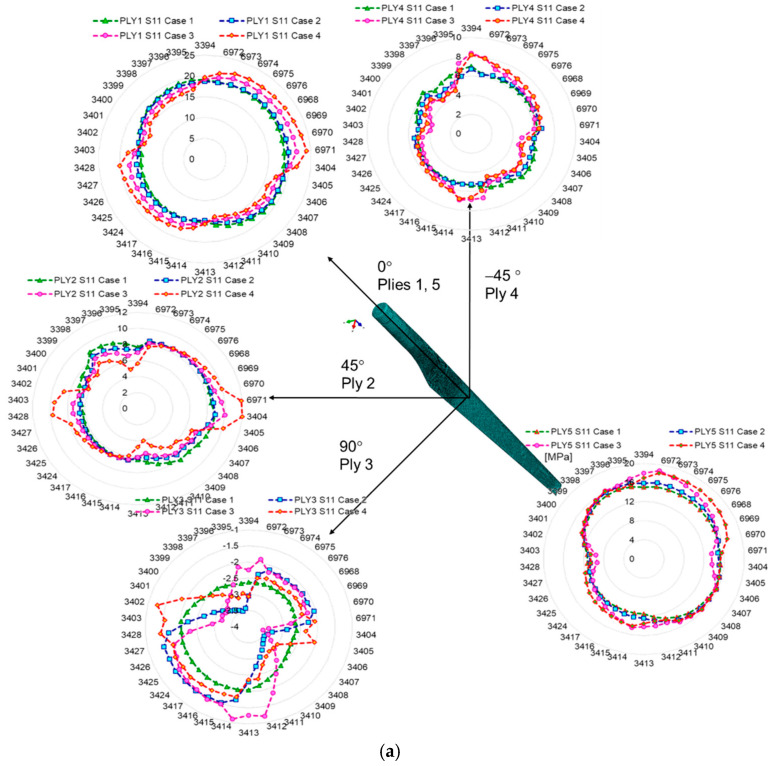
Variation of stresses in relation to the composite layer orientation, extracted from area 1: (**a**) S11 stress; (**b**) S22; (**c**) S12.

**Figure 7 polymers-14-01471-f007:**
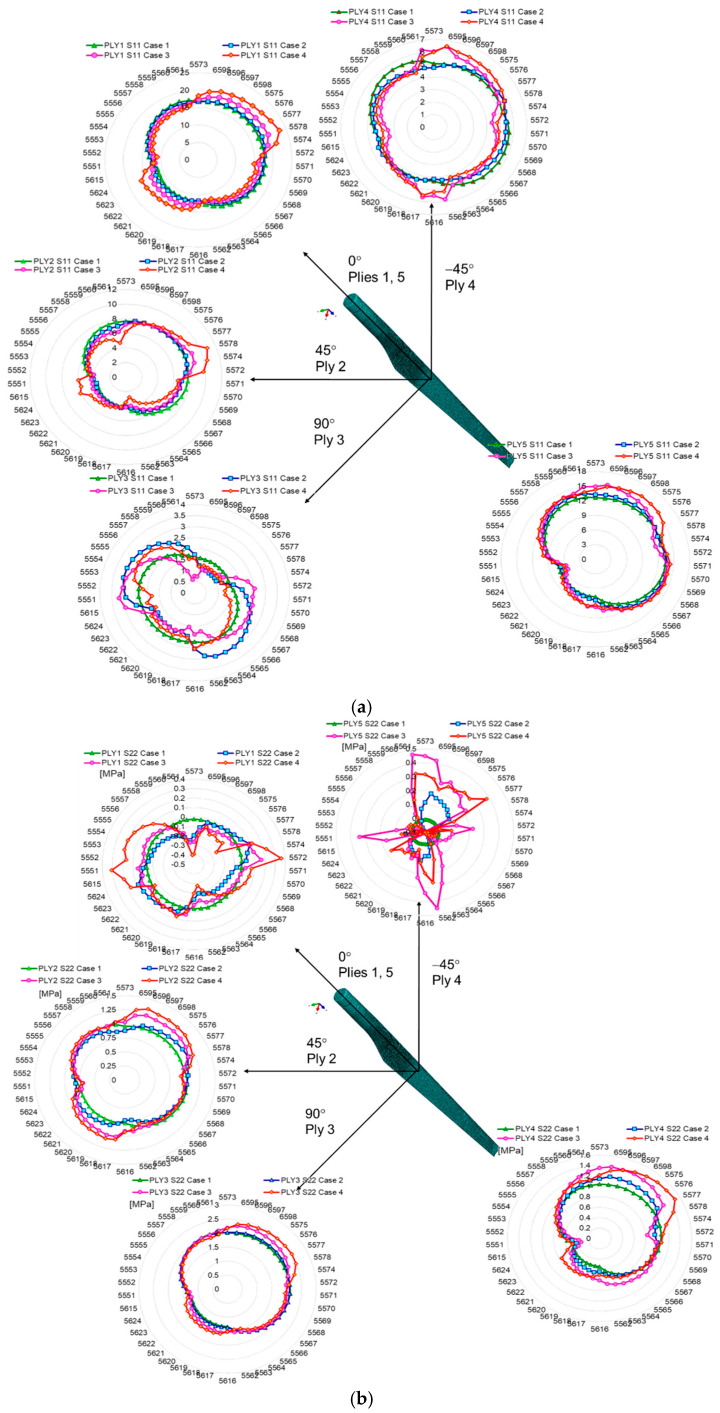
Variation of stresses in relation to the composite layer orientation, extracted from area 2: (**a**) S11 stress; (**b**) S22; (**c**) S12.

**Figure 8 polymers-14-01471-f008:**
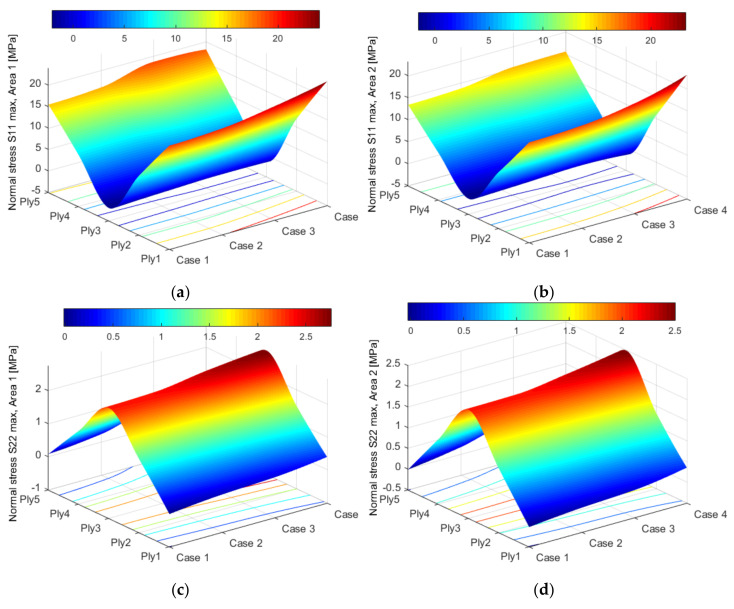
The relationship between the stress distribution and damage effects: (**a**) normal stresses S11 from area 1; (**b**) normal stresses S11 from area 2; (**c**) normal stresses S22 from area 1; (**d**) normal stresses S22 from area 2; (**e**) shear stresses S12 from area 1; (**f**) shear stresses S12 from area 2.

**Figure 9 polymers-14-01471-f009:**
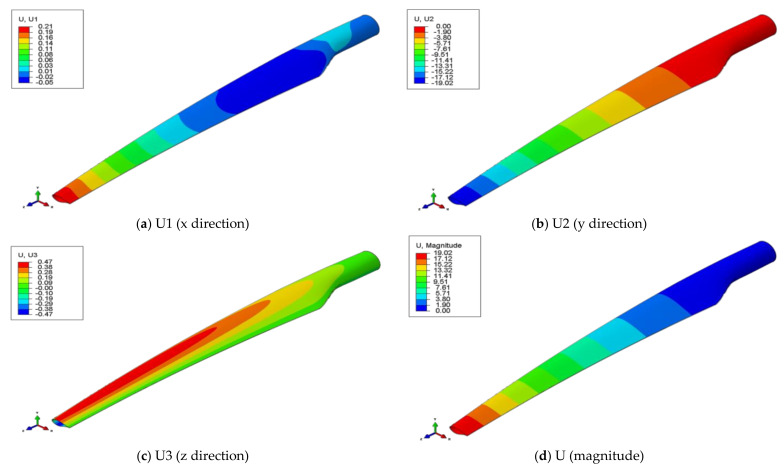
Distribution of displacement in case 2: (**a**) displacement in x direction; (**b**) displacement in y direction; (**c**) displacement in z direction; (**d**) total displacement.

**Table 1 polymers-14-01471-t001:** Elastic properties of each composite layer.

Composite RT500 Fiberglass Fabric Matrix: Epoxy Resin	Density ρ (kg/m^3^)	Thickness of Layer h (mm)	Number of Layers	Young’s Modulus (MPa)		Shear Modulus G_12_ (MPa)	Poisson Coefficient ν
				E_1_	E_2_		
0/45/90/45/0	2400	1.6	5	36,000	8800	3050	0.1615

**Table 2 polymers-14-01471-t002:** The displacement values for each studied case.

	Displacement (mm)				
Studied Cases	Total	%	On x	On y	On z
Case 1	18.96	0	0.21	18.96	0.47
Case 2	19.02	0.32	0.21	19.02	0.47
Case 3	19.10	0.74	0.22	19.10	0.48
Case 4	19.18	1.16	0.22	19.18	0.48

## Data Availability

Not applicable.

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
