# Peer review of "Prediction of the Damage Effect on Fiberglass-Reinforced Polymer Matrix Composites for Wind Turbine Blades"

_polymers, 2022, doi:10.3390/polym14071471_

Round 1
Reviewer 1 Report
The paper focuses on sensitivity analysis of effects of delamination degree of glass fibers reinforced polymers composites from wind turbine blade structure. The following major comments should be considered to further improve the quality of the paper.
*Abstract:
- In the abstract part, finite element analysis, are used to analyze the stress distribution and state wind turbine blades. However, how to determine the accuracy of finite element simulation. It is suggested to add relevant experimental verification results. In addition, the relationship between the stress distribution and damage effects should be determined.
* Introduction
- The introduction should be divided into several paragraphs, including research background, material information, properties and methods. The current writing lacks of some basic information of glass fiber reinforced polymer composites.
- The damage of wind turbine blades is not limited to lightning strike, which is also a probabilistic accident. In contrast, the long-term service of wind turbine blades in the actual harsh environment (high temperature and humidity, temperature alternation, loading effect, corrosion solution, etc.) is more likely to cause damage and rupture. For the wind turbine blades, they are generally composed of glass fiber reinforced polymer composites (GFRP). When considering damage evaluation and analysis, the summary of relevant literature on the damage (matrix cracking and the interface debonding and fracture (fiber fracture) of GFRP under long-term environment and loading should be added. Authors are advised to read the latest research as following:
Composite Structures, 2022. 281: 115060.
Construction and Building Materials, 2020, 245: 118399.
Journal of Materials Research and Technology, 2021, 14:2812-2831.
- According to the relevant work, the simulation of wind turbine blades has been done a lot. Therefore, where is the contribution and innovation of the present research work?
*Results and discussion
- All displayed results (Figure 5-12) are carried out through the finite element simulation, whether the authenticity of the results can be checked? It is suggested to add the experimental results of relevant literature to verify the accuracy of the finite element model.
- The results of finite element simulation should be displayed through the curve with the data points, such as the stress distribution along different directions. Through the curve, the readers can clearly observe the change trend of stress distribution.
- In addition, how to reflect the relationship between the simulation results and material damage?
Author Response
First we would like to thank the reviewers for carefully going through the manuscript and providing helpful suggestions for its improvement. Thanks to their constructive comments, we are able to present clearly and better version than the original manuscript. All the comments of the reviewers have been considered. In particular, the following changes have been made according to the reviewers' suggestions, highlighted by yellow color in the manuscript.

Reviewer 2 Report
There are some issues that require clarification.
In the introduction the authors identify the areas of study as at one-third and two-third points along the blade’s length, and cite refs 2-5. Ref 2 is in agreement with this. However, ref 3 identifies the trailing edge and a point at point at 0.73 of the radius from the centre (roughly in agreement with two-thirds), ref 4 identifies the leading edge and the tip, and ref 5 does not seem to identify any specific area. There does not seem to be a consensus as suggested by the citation of references 2-5. Would the authors wish to provide further support for their chosen damage areas?
Th FE calculations have been well executed. What is the justification of the end load of 600 N? Very detailed results of the effects of delamination are given.in Figs 6-11. The effects of delamination on the stresses are apparent. It is difficult to acquire an overall understanding of the effects from these figures. Figure 12 seems to be intended to provide this, but unfortunately its caption seems to have been left as the generic caption from the template. After fig.11 they state that “These results are in good agreement with mathematical models developed by [41 – 43].” It would much improve the paper if the authors could show a comparison to illustrate the level of agreement.
The conclusion is a restatement of the contents of the paper. It would be good to see some discussion of the usefulness of the modelling. Does it have any implications for blade design, materials choice, or the approach to modelling this system. We are left without much indication of the purpose or utility of the work.
Minor issues:
P1 line 30 “in operation (or in operation)” Not clear.
P2 line 86 Catia program. Needs a reference and description (a solid modelling program?)
P2 line 94. ABAQUS/Standard. Give version number and reference.
P3 Figure 2 Axis set is very small.
P6 Figure 5 An axis set is needed. This should show 1, 2, 3 directions. I assume these directions are aligned with the blade as suggested in Figure 1? Make this explicit so that we know exactly what S11 etc. refer to.
Author Response

(The authors gave the same response as above.)

Round 2
Reviewer 1 Report
The authors have responded well to all comments.